# Ionizing Radiation Sensing with Functionalized and Copper-Coated SWCNT/PMMA Thin Film Nanocomposites

**DOI:** 10.3390/nano13192653

**Published:** 2023-09-27

**Authors:** Guddi Suman, Merlyn Pulikkathara, Richard Wilkins, LaRico J. Treadwell

**Affiliations:** 1Sandia National Laboratories, Albuquerque, NM 87106, USA; gksuman@sandia.gov; 2Department of Chemistry and Physics, Prairie View A&M University, Prairie View, TX 77446, USA; mepulikkathara@pvamu.edu; 3Electrical and Computer Engineering Department, Prairie View A&M University, Prairie View, TX 77446, USA; rtwilkins@pvamu.edu

**Keywords:** N, N-Dimethylformamide, functionalized SWCNT/PMMA thin-film nanocomposites, X-ray, radiation sensor, COOH, copper-coated, ferrocene

## Abstract

This paper studies the ionizing radiation effects on functionalized single-walled carbon nanotube (SWCNT)/poly(methyl methacrylate) (PMMA) thin-film nanocomposites [SWNT/PMMA]. The functionalized thin-film devices are made of ferrocene-doped SWCNTs, SWCNTs functionalized with carboxylic acid (COOH), and SWCNTs coated/ modified with copper. The nanocomposite was synthesized by the solution blending method and the resulting nanocomposite was spin-cast on interdigitated electrodes (IDEs). A 160 kV X-ray source was used to irradiate the thin film and changes in the electrical resistance of the nanocomposites due to X-rays were measured using a semiconductor device analyzer. Carboxylic acid functionalized and copper-coated SWCNT/PMMA nanocomposite showed a reduced response to X-rays compared to unfunctionalized SWCNT/PMMA nanocomposite. Ferrocene-doped SWCNT showed a higher sensitivity to X-rays at lower dose rates. This is in contrast to a previous study that showed that similar nanocomposites using functionalized multi-walled CNTs (MWCNTs) had demonstrated an improved response to X-rays ionizing radiation compared to unfunctionalized MWCNTs for all dose rates. Electrical measurements were also performed using the Arduino Nano microcontroller. The result showed that a relatively economical, lightweight-designed prototype radiation sensor based on SWCNT/PMMA thin-film devices could be produced by interfacing the devices with a modest microcontroller. This work also shows that by encapsulating the SWCNT/PMMA thin-film device in a plastic container, the effect of ambient humidity can be reduced and the device can still be used to detect X-ray radiation. This study further shows that the sensitivity of SWCNT to X-rays was dependent on both the functionalization of the SWCNT and the dose rate.

## 1. Introduction

Carbon nanotubes (CNTs) are an allotrope of carbon, and they have exceptional electronic and mechanical properties [1,2]. Due to distinct properties such as a high aspect ratio, mechanical, electronic, and thermal properties, CNTs can be used for various applications, including radiation sensing for space and terrestrial applications. Carbon nanotubes have been studied as pressure, flow, thermal, gas, and chemical sensors [3,4,5]. Carbon nanotubes have also been found to be sensitive to low-dose ion radiation resulting in vacancies [6]. The covalent sidewall functionalization modifies the electronic band structure of SWCNTs that help to control the electronic properties of the CNTs giving band structure engineering for electronic and chemical sensor applications [7].

Recent advances in sensors based on CNTs show that they can be used for sensing biomolecules, gas, light, or pressure changes [8,9,10,11]. Unfunctionalized CNTs have chemically inert surfaces and a high specific surface area that provides adsorption sites for molecules to be adsorbed [8]. The charge transfer occurs between CNTs and adsorbed molecules, which increases or decreases the conductivity of CNTs. The functionalized carbon nanotubes with polymers [8,12,13,14], functional groups [13,15,16], and metal nanoparticles [9,13] enhance the response of the sensor due to changes in their electronic properties. Thin films have been a way of creating sensors and have been studied in the literature, for example; the thin films of the type CuX (X = Bi, Mg, Ni) studied the impact of the concentration of copper and phases containing copper on the degradation and oxidation of thin films [17]. Another example of synthesis of NiFe films using direct, pulse, and pulse-reverse electrodeposition modes studied the deposit composition, crystal structure, and surface microstructure [18]. There has been mechanical modeling of thin films [19,20], PMMA [21] and a polymer/carbon nanotube [22]. The experimental thermomechanical property was studied on materials such as ceramics [23], PMMA/SWCNT interfaces [24], and PMMA and modified SWCNT [25]. This work focuses on electrical properties of thin film nanocomposites of PMMA/nanotubes. In previous work, unfunctionalized MWCNTs and functionalized MWCNT nanocomposites were studied as ionizing radiation sensors [26,27]. X-ray radiation effects on thin-film nanocomposites of functionalized nanotubes with carboxylic acid, hydrogen peroxide, amine, ferrocene-doped, and copper-coated multi-walled carbon nanotubes, and poly (methyl methacrylate) (PMMA), that is (SWNT/PMMA), were studied in ref. [27]. The functionalized MWCNT nanocomposites were found to be more sensitive than the unfunctionalized MWCNTs. The sensitivity of the thin film devices, as measured in a decrease in resistance of the devices, of ferrocene-doped MWCNT/PMMA nanocomposites was about three times greater in comparison to unfunctionalized MWCNT/PMMA devices due to incident X-ray radiation [27]. The presence of functional groups on MWCNTs caused a higher sensitivity to X-rays than unfunctionalized MWCNTs/PMMA [26]. The functional groups with MWCNTs provide an additional source of excess charge carriers, causing enhanced sensitivity of MWCNTs to X-rays. The functionalized carbon nanotubes have electronic, mechanical, and optical properties which are different from pristine nanotubes and can be useful for sensing applications [12]. In related work, highly sensitive real-time proton dosimeters functionalized SWCNTs were fabricated, such as fluorinated SWCNT (F-SWCNTs), palladium doped SWCNTs (Pd-SWCNTs), and nanotubes non-covalently associated with cellulose (Cel-SWCNTs) were studied as proton detectors [28]. The result showed that they have good lifetime performance and can be used repeatedly. 

Unfunctionalized MWCNT, functionalized MWCNT, and unfunctionalized SWCNT thin films showed a directly proportionate response to applied X-ray doses and dose rates [26,27,29]. The functionalized MWCNT thin film showed a higher response to X-rays than unfunctionalized MWCNTs. The present work presents the response of SWCNT functionalized nanocomposites with carboxylic acid, copper-coated SWCNTs, and ferrocene-doped SWCNTs to X-rays. The SWCNT/PMMA nanocomposite thin films were irradiated with X-rays at different dose rates to a total dose of 100 krad. The observed results were compared with previous work with devices using unfunctionalized SWCNT/PMMA thin films [29]. As observed in previous work, a decrease in the resistance of the thin film was measured during X-ray exposure with the higher response of ferrocene-doped SWCNT/PMMA film at a lower X-rays dose rate as compared with unfunctionalized SWCNT/PMMA films. In contrast to the MWCNT work, SWCNT/PMMA films composed of copper-coated and carboxylic acid functionalized SWCNT nanocomposites showed a lower response to X-rays than unfunctionalized SWCNT/PMMA film. 

The findings in this study showed that functionalization is not as beneficial for SWCNTs as compared to MWCNTs, which is an important consideration in developing these nanocomposites for practical ionizing radiation sensors. Other practical aspects of using these thin film nanocomposite devices to detect radiation were studied in this work. The preliminary groundwork for producing a prototype instrument based on SWCNT/PMMA thin-film nanocomposites and a common microcontroller was completed. The thin-film device was interfaced with the Arduino Nano board with a supported microcontroller to monitor the decrease in resistance of the thin film during radiation exposure. In addition, the thin-film device was encapsulated to help isolate the device from the external ambient environment. The results showed that by encapsulating the device in a plastic container, the effect of humidity could be reduced and the device could still be used to detect X-ray radiation.

## 2. Materials and Methods

### 2.1. Materials for the Fabrication of the Nanocomposite

The commercial unfunctionalized SWCNT, carboxylic acid-functionalized SWCNT (D1.5L1-5-COOH), and copper-coated SWCNT (custom ordered) are produced by a thermal chemical vapor deposition method with greater than 95% purity, having a length of 1–5 µm and a diameter of approximately 1.5 nm. These were purchased from NanoLab, Inc., Waltham, MA, USA). Ferrocene, an organometallic compound, N, N-Dimethylformamide (DMF) (319937)), and PMMA (182230) were purchased from Sigma Aldrich. PMMA was chosen as the polymer for fabricating thin films in this study because of its good spinning qualities [30], solubility in DMF [29], its amorphous character (thereby avoiding crystallization during the cooling process) [31], high weatherability (durability) [32], and good chemical resistance [33]. While X-rays are known to damage PMMA [34] more than other polymers, PMMA was chosen for our work to facilitate device fabrication for the reasons given above. Future research may explore the utilization of other more radiation-tolerant polymers for the nanocomposite.

The use of these particular modified SWCNTs was due to their ready availability as they were a subset of the MWCNTs used in ref. [27]. Interdigitated electrodes (IDEs), which were patterned on a silicon substrate, upon which the nanocomposite thin films are cast and used to measure the film resistance, were purchased from Case Western Reserve University. Silicon substrates with an IDE pattern are 8 mm × 8 mm, 0.3 mm thick, and have a 300 nm thick silicon dioxide layer on top. The electrodes (with a line size and spacing of 100 µm) and bonding pads (1.75 mm × 1.75 mm) patterned on top of the substrates are made of 200 nm-thick gold layers on top of a 10 nm-thick titanium adhesion layer, shown in Figure 1(Ai). Figure 1(Aii) shows a scanning electron microscope (SEM) image of a ferrocene-doped SWCNT/PMMA thin-film nanocomposite deposited on an IDE.

### 2.2. Fourier Transform-Infrared (FTIR) Spectroscopy

The FTIR spectroscopy analysis was carried out on the prepared samples with a Thermo Scientific Nicolet iS50 FTIR spectrometer to identify the functional groups present in the thin film composites and their interaction mechanism with SWCNT and PMMA. PMMA, unfunctionalized SWCNT/PMMA, ferrocene-doped SWCNT/PMMA, copper-coated SWCNT/PMMA, carboxylic acid-functionalized SWCNT/PMMA thin film composites and unfunctionalized SWCNT/PMMA composite were recorded in the wavelength region of 1800–600 cm^−1^ as shown in Figure 1B. 

PMMA powder shows C–H stretching around 650–1000 cm^−1^, C–O stretching from 1060–1265 cm^−1^ which is in the range 1050–1260 cm^−1^ [35], and C=O stretching at 1715 cm^−1^ [35]. Unfunctionalized SWCNT/PMMA, copper-coated, ferrocene-doped and carboxylic acid functionalized SWCNT/PMMA composite showed the presence of C–O and C=O stretching due to PMMA as the concentration of PMMA was higher than functionalized SWCNT (SWCNT with functional groups/PMMA = 1:50 by weight). In the case of the ferrocene-doped SWCNT/PMMA composite, ferrocene was embedded into PMMA(ferrocene: PMMA = 1:100 by weight) and the composite shows the presence of a weak band due to PMMA and the absence of C–C stretching (in-ring) from 1400–1500 cm^−1^ and 1585–1600 cm^−1^ due to the ferrocene [35]. There is a difference in IR spectra of unfunctionalized SWCNT/PMMA and copper-coated SWCNT/PMMA composites, which may be due to the presence of copper-coated SWCNT in the composite. The band around 1690–1760 cm^−1^ in carboxylic acid-functionalized SWCNT/PMMA composite corresponds to a carbonyl stretch C=O of carboxylic acid [25,35].

### 2.3. Functionalized SWCNT/PMMA Nanocomposite Thin Film Preparation Method

The functionalized SWCNT/PMMA nanocomposite thin film (carboxylic acid, copper-coated) was prepared in the same manner as the unfunctionalized SWCNT/PMMA film [29] and the MWCNT nanocomposites were studied in ref. [27]. The PMMA dissolved in DMF with a concentration of 0.03 g/cm^3^ was tip-sonicated for 15 min at 20% amplitude using 750 W, a 20 kHz probe sonicator at a pulse rate of 0.5. In this study, the loading of CNT was selected to ensure the consistent resistance of the thin-film devices. The resistance range of the devices was chosen for convenience of measurement; therefore, a maximum loading of CNT was not determined in our work. 

Functionalized copper-coated SWCNT in DMF with a concentration of 2.29 mg/cm^3^ was added to the above sonicated mixture of PMMA/DMF and was sonicated for 8 min. The sonicated mixture of copper-coated SWCNT/PMMA was deposited on IDEs using a spin coater at 1000 rpm at 100 rpm/s for 30 s to create a thin film of nanocomposite with a network of nanotubes spanning the electrodes and bridging IDEs fingers, Figure 1(Ai). After the thin films were cast on IDEs, they were cured in a vacuum oven at 175 °C for 1 hour. The resulting devices were taken out of the oven and left for cooling overnight to allow the excess solvent to evaporate from the nanocomposite thin film. Carboxylic acid functionalized SWCNT/PMMA nanocomposite thin film was prepared similarly with a variation in the concentration of carboxylic acid functionalized SWCNT and time for sonication. The variation in concentration was needed to ensure consistency of electrical resistance over all the devices. Carboxylic acid functionalized SWCNTs in DMF with a concentration of 6.03 mg/cm^3^ was added to the sonicated mixture of PMMA in DMF and sonicated again for 7 min. 

Ferrocene-doped SWCNT/PMMA nanocomposite thin films were prepared by a similar method as the unfunctionalized SWCNT/PMMA thin film [29], only with a slight variation in the process of sonication of PMMA in DMF. The ferrocene in the DMF with a concentration of 2.62 mg/cm^3^ was added to a vial containing DMF and then PMMA was added to a vial containing ferrocene and DMF (concentration of PMMA in DMF was 0.03 g/cm^3^) and the mixture of ferrocene, PMMA, and DMF was tip-sonicated in the presence of ice for 15 min. Figure 1(Ai,Aii) shows an SEM image of IDE and ferrocene-doped SWCNT/PMMA nanocomposite thin film deposited on the IDE pattern of the silicon substrate. Homogeneity and adequate dispersion of the nanocomposite samples were confirmed by the SEM image of the ferrocene doped-SWCNT/PMMA, as shown in Figure 1A, and the EDS of the copper-coated nanotubes are shown in the color mapping, which shows the distribution in Appendix A and of the ferrocene doped-SWCNT/PMMA in Appendix A.

## 3. Experimental Setup

A 160 kV, 3000 W X-ray source (X-RAD iR-160) was used to irradiate the thin-film nanocomposite devices with X-ray radiation. An X-ray tube in which an energetic electron beam impinges on a tungsten target produced an X-ray spectrum with sharply defined peaks at 55.55 keV and 66.66 keV with a low-lying broad bremsstrahlung background. The resulting X-ray beam has a uniform circular cross-section that is about 15.24 cm in diameter at the surface of the tray in which the thin-film device was positioned in the center of the tray. The dose rate effect on the resistance of these film devices: the dose rate is defined by changing the distance of the thin-film device from the source of the X-rays. For example, a dose rate of 5.92 krad/min, 2.13 krad/min, and 1.08 krad/min is achieved by placing the thin-film device at a distance of 30 cm, 50 cm, and 70 cm, respectively, from the center of the X-ray source. To perform the X-ray dose rate experiment, the nanocomposite thin-film device was placed inside the X-ray chamber. The device was placed in the X-ray chamber to insure uniform irradiation at the prescribed dose rate. The device was connected to source/measure units (SMUs) of a Keysight B1500A semiconductor device analyzer using a long, low noise cable. The control device was placed outside the X-ray machine and monitored concurrently with the irradiated device. The semiconductor device analyzer was programmed to apply a voltage and measure the current through the device every second. The data were collected simultaneously for pre-irradiation, during irradiation, and post-irradiation for the irradiated and control devices. The schematic diagram of the experiment is shown below in Figure 2A.

Previously, a CNT-based electrochemical biosensor and pressure sensor was studied where electrical measurements were performed using an Arduino [36,37]. While most data presented in this work were taken using a semiconductor parametric analyzer described below, the thin-film devices were also interfaced with an Arduino Nano board. The Arduino Nano board is a small, flexible microcontroller board with microcontroller ATmega328P developed by Arduino, as shown in Figure 2B. The microcontroller board operates at 5 V and has 14 digital and 8 analog pins. These analog and digital pins can be configured as input and output. Analog pins have a total resolution of 10 bits which measure the value from 0 to 5V. Arduino 1.8.19 software was installed on a laptop computer. The source code to measure the change in resistance of the sample during radiation exposure was written on Arduino software downloaded on the computer and the software communicated with the Arduino Nano board through a Mini-B-USB [38]. We repeated the radiation experiment on a nanocomposite thin-film device using an Arduino Nano board, described below.

In this study, a thin-film nanocomposite device was interfaced with an Arduino Nano microcontroller board, as shown in Figure 2(Bd). An Arduino Nano platform was programmed to measure the resistance of the thin-film device. The circuit implemented to measure the change in resistance was a simple voltage divider, where the nanocomposite device was connected in series with a reference resistor, and the change in resistance of the nanocomposite device was measured during irradiation by monitoring the voltage across the reference resistor.

An Arduino microcontroller (Figure 2(Ba)) was wired with a DHT11 temperature/humidity sensor (Figure 2(Bb)) to collect the temperature and humidity readings during the experiment. DHT11 has a humidity sensing component that is a substrate that can hold moisture with electrodes connected to the surface. The substrate releases ions when it absorbs water vapor resulting in an increase in conductivity between electrodes. It has an NTC temperature sensor (thermistor) to collect temperature data [39].

## 4. Results

The radiation effect on the SWCNT/PMMA thin film due to X-ray radiation was studied through changes in the resistance of the thin films such that the thin film acts as the sensing material. The following expression shows the change in resistance (S) of the SWCNT/PMMA thin films due to radiation in this work: $$S=\left(\frac{R_{f}-R_{i}}{R_{i}}\right)\times 100\%$$

R_f_ indicates the final resistance of the thin film after radiation exposure at fixed-dose and dose-rate conditions and R_i_ is the initial resistance before radiation exposure. The resistance of the nanocomposite thin films is measured by a semiconductor device analyzer (or the microcontroller) as a function of time at each second during a radiation experiment. 

### 4.1. Dose Rate Response

The change in electrical resistance of a ferrocene-doped SWCNT/PMMA nanocomposite thin-film device, when irradiated to different X-ray dose rates 1.08 krad/min, 2.13 krad/min, and 5.92 krad/min at a total dose of 100 krad, is shown in Figure 3. The results of this experiment show the continuous decrease in resistance of the device during the radiation exposure time and a prolonged decrease in resistance after radiation exposure. Because the polymer is an insulator, the electrons take some time to drain from the polymer, which is why the resistance continues to decrease after the irradiation stops. The magnitude of the decrease in resistance of the thin film device was inversely proportional to the dose rates.

When these experiments were repeated with copper-coated and functionalized carboxylic acid SWCNT/PMMA nanocomposite thin-film devices as shown in Figure 4 and Figure 5, similar responses were observed with significant differences in the magnitude of the decrease in device resistance in comparison to ferrocene-doped SWCNT/PMMA at different dose rates.

Figure 6 shows the results of the dose rate experiment of functionalized and unfunctionalized SWCNT/PMMA thin-film devices at different dose rates to obtain a total dose of 100 krad X-rays. Each device was irradiated to 100 krad at different dose rates of 5.92 krad/min, 2.13 krad/min, and 1.08 krad/min, respectively. The decrease in resistance of the ferrocene-doped SWCNT/PMMA thin-film device was quite similar to unfunctionalized (Un-func) at the highest dose rate of 5.92 krad/min as shown in Figure 6.

However, the magnitude of decrease in resistance of ferrocene-doped SWCNT/PMMA was highest of all the other functionalized and unfunctionalized SWCNT/PMMA thin-film devices at lower dose rates of 2.13 krad/min and 1.08 krad/min. Figure 6 shows that the magnitude of decrease in the resistance of carboxylic acid-functionalized and copper-coated SWCNT/PMMA was significantly smaller than the ferrocene-doped and unfunctionalized SWCNT/PMMA for all dose rate conditions, with almost a factor of two. Six devices of each functionalization were subjected to radiation exposure, and similar results were observed for all devices. These experiments were repeated for all the devices, and the vertical bars show standard deviation. These results show the reproducibility of the experimental results. 

### 4.2. Thermal Treatment and Reset

When the thin film nanocomposite device was irradiated with X-rays, the thin film’s resistance decreased, as shown in Figure 3, Figure 4 and Figure 5. Unfunctionalized MWCNTs, and functionalized MWCNTs and unfunctionalized SWCNTs showed similar results [26,27,29] for the dose rate experiment. The nanocomposite device was reset by heat treatment at 175 °C in 24 min after X-ray exposure of a total dose of 100 krad at a dose rate of 1.08 krad/min. Similar heat recovery responses of ferrocene-doped and copper-coated SWCNT/PMMA were observed. Figure 7 shows the heat treatment data of the carboxylic acid functionalized SWCNT/PMMA nanocomposite.

Initial average resistances and changes in resistance of fabricated devices due to total 100 krad X-ray doses at dose rate of 5.92 krad/min are mentioned in Table 1.

### 4.3. Steps towards a Practical Radiation Detection Instrument

The dose-rate experiment was performed on the unfunctionalized SWCNT/PMMA device using a semiconductor device analyzer when the nanocomposite device was irradiated to X-rays at a total dose of 100 krad at a dose rate of 5.92 krad/min. The same previously irradiated nanocomposite device, after a thermal reset, was connected to the Arduino Nano and irradiated to X-ray radiation at a total dose of 100 krad at the dose rate of 5.92 krad/min. The decrease in resistance during radiation exposure was recorded with the results shown in Figure 8. The reduction in resistance of the thin-film nanocomposite device in both cases was nearly the same during irradiation and varied somewhat (about 1%) during post-irradiation measurements. These results indicate that the SWCNT/PMMA nanocomposite device interfaced with the Arduino Nano board, or a similar microcontroller, may be used to build a lightweight radiation sensor instrument. This experiment was repeated for about ten devices and similar results were obtained.

The response of the unfunctionalized SWCNT/PMMA thin-film device to ambient humidity was studied. This study is essential for developing practical radiation detection instruments for terrestrial field measurements. The data were collected using the Arduino Nano microcontroller to prototype a lightweight, prototype radiation sensing device. It is necessary to understand and mitigate the environmental humidity effect on the thin-film device for radiation sensing application. The typical results when a ferrocene-doped SWCNT-PMMA thin-film device under open and encapsulated (sealed) condition to the ambient environment is shown below in Figure 9. Together with changes in the device resistance, ambient temperature and humidity data are collected simultaneously. The change in resistance of the device under open condition without encapsulation was about −1% and with encapsulation was approximately less than −0.1%. The experiment was repeated with the unfunctionalized, copper-coated and carboxylic acid functionalized SWCNT/PMMA thin-film device and similar results were observed. 

To further study the effects of encapsulation, unfunctionalized SWCNT/PMMA, ferrocene-doped SWCNT/PMMA, copper-coated SWCNT/PMMA, and carboxylic acid (COOH) functionalized SWCNT/PMMA thin-film devices under open conditions were exposed to 100 krad X-rays at 5.92 krad/min and the decrease in resistance of the devices due to X-ray radiation was monitored. All the devices were reset, encapsulated, and again irradiated with 100 krad X-rays at 5.92 krad/min and the decrease in resistance of the devices due to X-rays was monitored; the result is shown in Figure 10. Figure 10 shows that the encapsulation of the device reduces the amount of decrease in resistance of the device due to X-rays at a maximum of 4%. The result showed that even encapsulated thin-film devices could be used to detect X-rays radiation with energies produced by the source. It can be observed that the highest decrease in resistance during X-ray exposure for the encapsulated device was about 18%. However, it should be noted that similar experiments that resulted in the data of Figure 9 and Figure 10 were also repeated for functionalized SWCNT/PMMA devices with similar results.

The experiments reported here have not focused on the aging effects on the devices. However, we used devices over a period of at least 1.5 years without substantial loss of sensitivity to X-rays. In the low-dose-rate environments that exist in space and some other terrestrial environments, the nanocomposite devices may be suitable for long-term sensor applications.

## 5. Discussion

The decrease in thin film resistance was inversely proportional to the applied X-ray dose rates, as shown in Figure 3, Figure 4 and Figure 5. X-rays caused an ionization effect both in SWCNT and PMMA, which generated radiation-induced excess charge carriers, an increase in the conductivity and decrease in the resistance of the nanocomposite film [26,27,32,33] with the slow release of charges in the insulating polymer near the CNT-polymer interface. The radiation-induced charged carriers are transported to electrodes of the IDEs through the networking of nanotubes between the electrodes. Electrons transport contributes to conductivity, and holes are more likely to become trapped in the polymer matrix and the oxide layer of the IDE. This could explain the decrease in the resistance of the nanocomposite thin-film device during X-ray exposure and the slow recovery of the nanocomposite after the X-rays have been turned off [27,29]. It should be noted that devices made from carbon black with similar initial device resistance showed no significant response to the same X-ray environment indicating the importance of the nanotube network within the thin films [26]. It is noted in the data that there are instances where the resistance of the samples continues to decrease even after the X-rays were turned off (for example, the data in Figure 3 for each dose rate). This is likely due to the relatively slow drift conduction to the CNT network of charge carriers generated in the polymer matrix (a poor conductor) during irradiation. Some charges likely become trapped and the thermal treatment discussed in Section 4.2 provides sufficient thermal energy to these trapped charges in the IDE oxide and polymer matrix so that the sample resistance relaxes to a pre-irradiation condition in a relatively short time [27].

The increased rate in the change of resistance of the thin film at higher dose rates, as shown in Figure 3 and Figure 5, is caused by the larger electron and hole pair production. The large electron-hole pair production in the thin film during X-ray exposure also results in the formation of defect sites [40,41] along the sidewalls of the nanotubes. However, Figure 4, for the nanocomposite with copper-coated nanotubes, shows that the higher dose rate caused a change in resistance slower compared to the other two dose rates. There may be other influences due to the presence of copper-coating on SWCNTs. A previous study mentioned that SWCNTs cause a reduction in oxidation of the copper affecting the electrical conductivity of the copper-coated SWCNT/PMMA nanocomposite [42] and this may indicate the possibility of a slower rate of change in the resistance of copper-coated SWCNT/PMMA at higher dose rates of X-ray exposure. Another study mentioned the X-ray photoelectron spectroscopy (XPS) for elemental analysis of Cu-SWCNT composites produced by electroless plating. XPS detects the presence of copper (II) hydroxide and cuprous oxide. The reaction of copper oxide and atmospheric moisture may have caused the presence of Cu (OH)_2_ [42]. The complex transition metal compounds easily cause the oxygen excess and/or deficit [20,43]. In addition, the PMMA matrix itself has a stoichiometric formula (C_5_O_2_H_8_)n, and it breaks down to monomers at 165^0^C causing oxygen diffusion [44,45]. The decrease in resistance of the ferrocene-doped SWCNT was significantly greater than all other types of SWCNT studied at the lowest dose rate, as shown in Figure 6. The presence of ferrocene groups with SWCNT/PMMA increased the sensitivity to X-rays at lower dose rates, even in comparison with unfunctionalized, COOH functionalized and copper-coated SWCNT/PMMA devices. The bonds connecting ferrocene and CNT are broken during the X-ray exposure, which generated additional charge carriers for conduction [26,46]. The change in resistance of copper-coated and carboxylic acid functionalized SWCNT/PMMA is significantly less than the unfunctionalized SWCNT/PMMA and ferrocene-doped SWCNT/PMMA thin-film devices. Thus, the pristine nature of the unfunctionalized SWCNT outweighs any benefit of additional charge carriers for the COOH and copper-coated SWNT nanocomposite sites. For higher dose rates, the ferrocene-doped and unfunctionalized devices were the most sensitive, and for lower dose rates, the ferrocene-doped devices were the most sensitive. This observation would be critical in optimizing radiation sensors based on these CNT/PMMA nanocomposite thin films for any radiation detection instrument. 

The functional group attached to CNTs causes modification on the electronic properties of nanotubes which is independent of the type of functional groups and the variation in the C–C bonding structure from the *sp*^2^ bonding configuration. However, the thin-film devices consisting of functionalized MWCNTs showed greater sensitivity to X-rays than devices consisting of unfunctionalized MWCNTs [25]. The functional group attached to CNTs also results in the generation of an impurity state in the gap region. The impurity state acts as a scattering center for the conduction carriers around the Fermi level, and thus changes the ballistic transport of metallic nanotubes, electronic states, and conducting properties of CNTs [7]. The effect of the reduction in conductivity in SWCNT due to functionalization may be more in SWCNT/PMMA nanocomposites than MWCNT/PMMA due to differences in their morphological structures. For the low-dose radiation detection application, the SWCNT/PMMA thin-film device showed a stepwise electrical resistance due to ambient humidity and temperature. However, when the device was encapsulated within an airtight plastic container, the effect of ambient conditions was mitigated, resulting in the stable electrical resistance of the device, as shown in Figure 9. The airtight plastic container not only controlled the environmental humidity but also allowed the device to detect X-ray radiation, as shown in Figure 10. Further experiments with a larger airtight plastic container that could hold a DHT11 temperature and humidity sensor would show the stable conditions within the container.

## 6. Conclusions

This study performed dose-rate experiments on functionalized SWCNT/PMMA thin-film devices that were performed by irradiating the thin-film devices with X-ray radiation at different dose rates of 5.92 krad/min, 2.13 krad/min, and 1.08 krad/min to the total dose of 100 krad. The results showed that ferrocene-doped and unfunctionalized SWCNT/PMMA devices were more sensitive to X-rays compared with the devices containing copper-coated or COOH functional groups. The addition of ferrocene groups increased the sensitivity of the SWCNT/PMMA nanocomposite at lower X-ray dose rates. From an engineering point of view, the ferrocene-doped SWCNT/PMMA thin-film device may be best for the actual detector system, at least for lower dose rate applications such as the aerospace environment. The thin-film nanocomposite device can be heat-treated to recover the initial resistance. The findings in this study also showed that the environmental humidity effect on thin-film devices could be reduced by encapsulating the device and the device can still be used to detect radiation. The response of thin-film SWCNT/PMMA nanocomposites monitored by a semiconductor device analyzer and Arduino Nano produced similar results during irradiation. While device aging was not explicitly studied here, our experiments showed that the nanocomposite devices remained tolerant to X-ray exposures up to 1000 krad and that the devices remained functional for at least 1.5 years. Future work may include exploring other functionalized SWCNT and other nano-scale materials such as graphene together with more radiation-tolerant polymers to produce a lightweight, portable sensor device with high sensitivity for X-ray studies for space and terrestrial applications.

## Figures and Tables

**Figure 1 nanomaterials-13-02653-f001:**
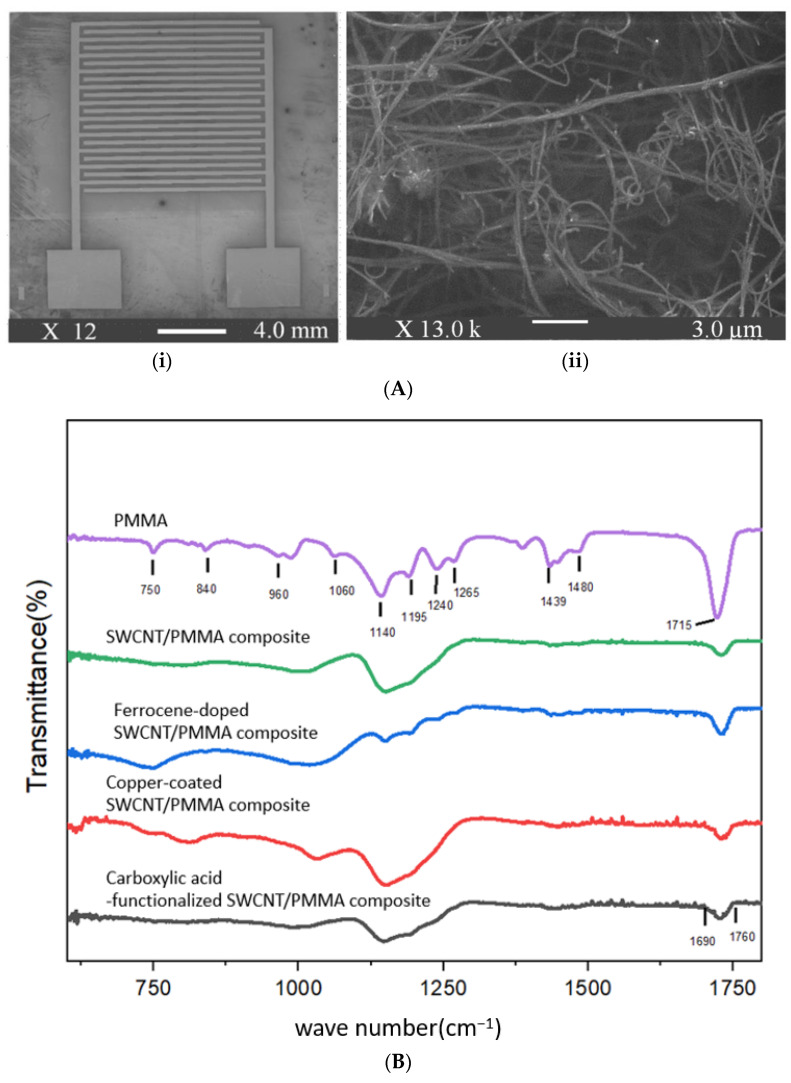
(**A**) SEM image of (**i**) interdigitated electrode, and (**ii**) ferrocene-doped SWCNT/PMMA nanocomposite film. (**B**) FTIR spectrum of PMMA, unfunctionalized SWCNT/PMMA, ferrocene-doped SWCNT/PMMA, copper-coated SWCNT/PMMA, carboxylic acid-functionalized SWCNT/PMMA thin film composites and unfunctionalized SWCNT/PMMA composite.

**Figure 2 nanomaterials-13-02653-f002:**
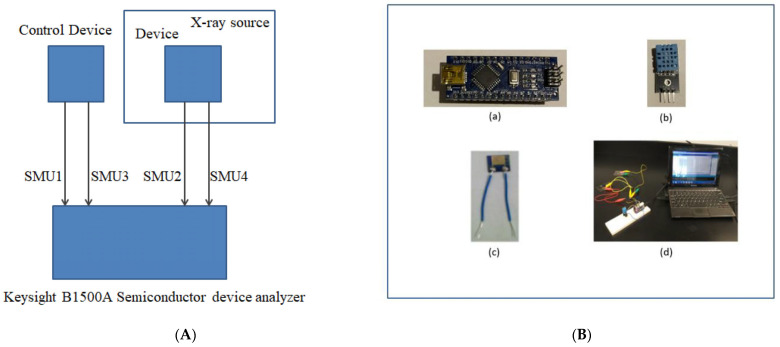
(**A**) The schematic diagram of the experiment. (**B**) (**a**) Arduino board (picture of board taken in experimental set up), (**b**) DHT11 Temperature & Humidity Sensor (picture of DHT11 taken in experimental setup), (**c**) thin film device, and (**d**) experimental setup using Arduino Nano board and DHT11 connected to thin-film device.

**Figure 3 nanomaterials-13-02653-f003:**
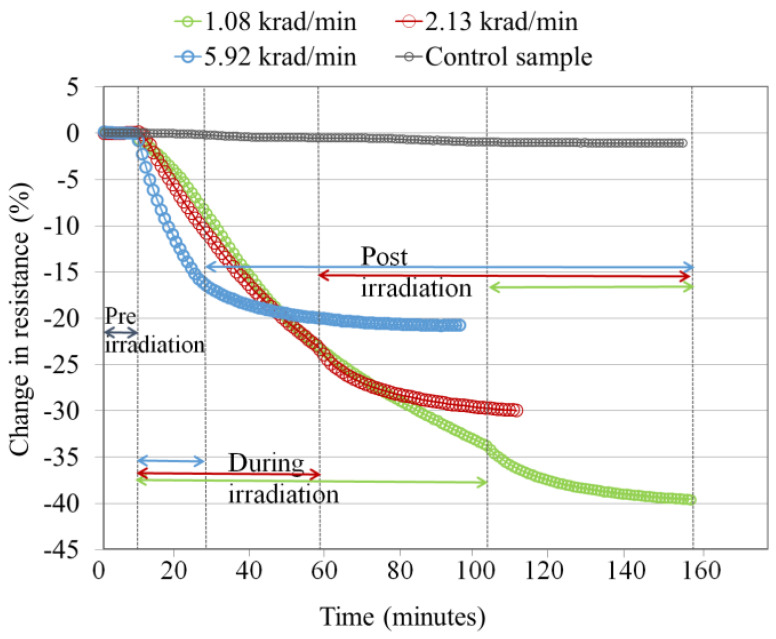
Ferrocene-doped SWCNT/PMMA nanocomposite irradiated to a total dose of 100 krad at different dose rates. The color of the data points is coordinated with the dose rates and time sequence of the exposures for clarity.

**Figure 4 nanomaterials-13-02653-f004:**
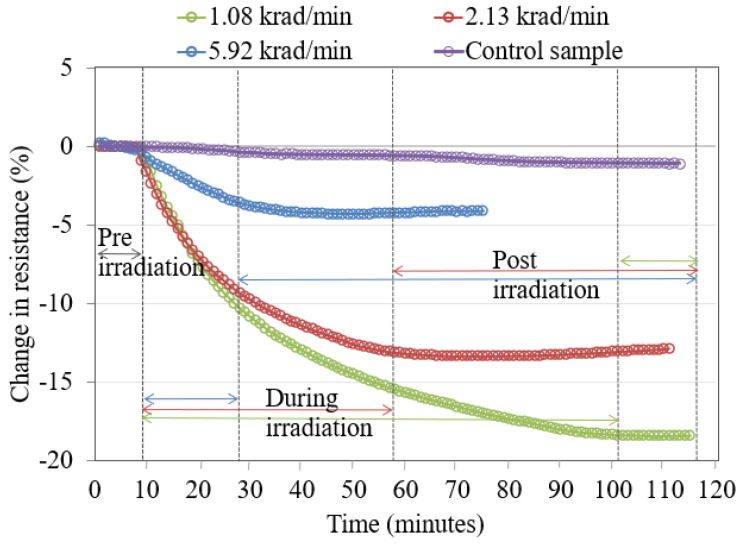
Copper-coated SWCNT/PMMA nanocomposite irradiated to a total dose of 100 krad at different dose rates.

**Figure 5 nanomaterials-13-02653-f005:**
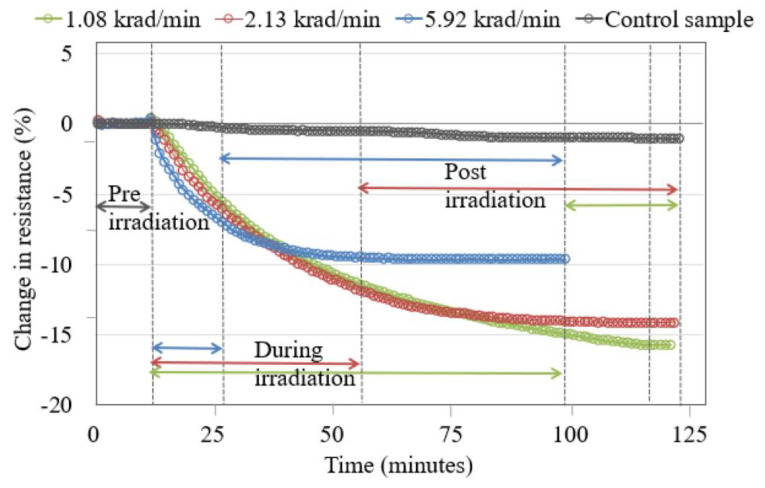
Carboxylic acid-functionalized SWCNT/PMMA nanocomposite irradiated to a total dose of 100 krad X-rays at different dose rates.

**Figure 6 nanomaterials-13-02653-f006:**
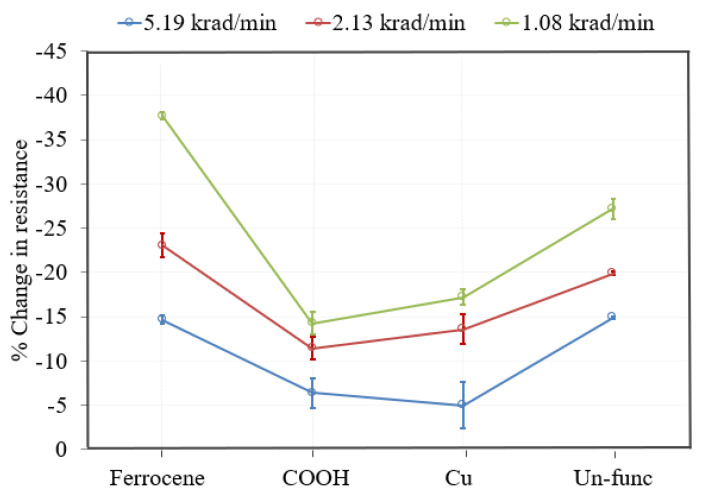
Responses of unfunctionalized [Un-func] and functionalized [Ferrocene, COOH, Cu] SWCNT/PMMA at different dose rates to a total dose of 100 krad.

**Figure 7 nanomaterials-13-02653-f007:**
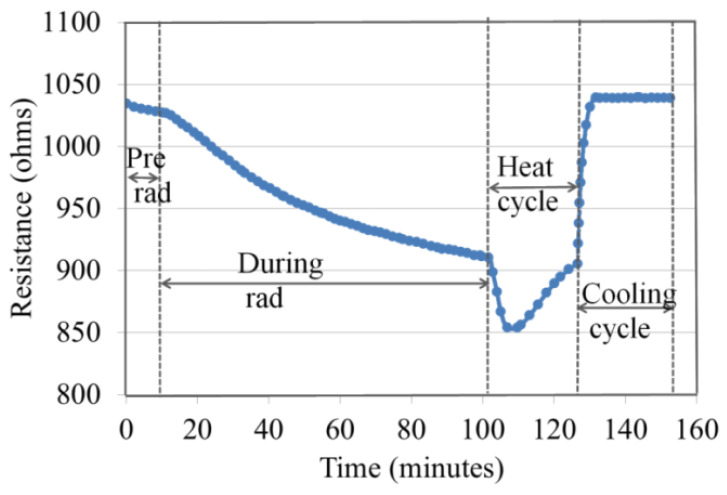
Heat reset cycle of carboxylic acid functionalized SWCNT/PMMA nanocomposite after being irradiated to a total dose of 100 krad at a dose rate of 1.08 krad/min.

**Figure 8 nanomaterials-13-02653-f008:**
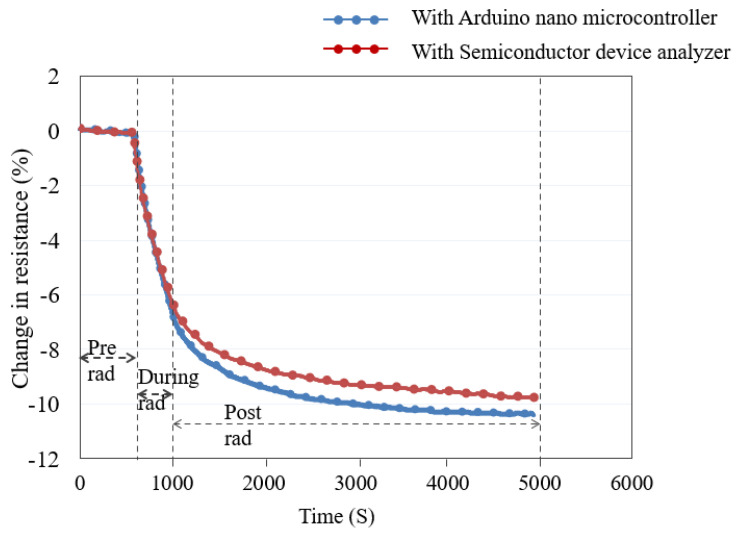
SWCNT/PMMA device integrated with Arduino Nano microcontroller.

**Figure 9 nanomaterials-13-02653-f009:**
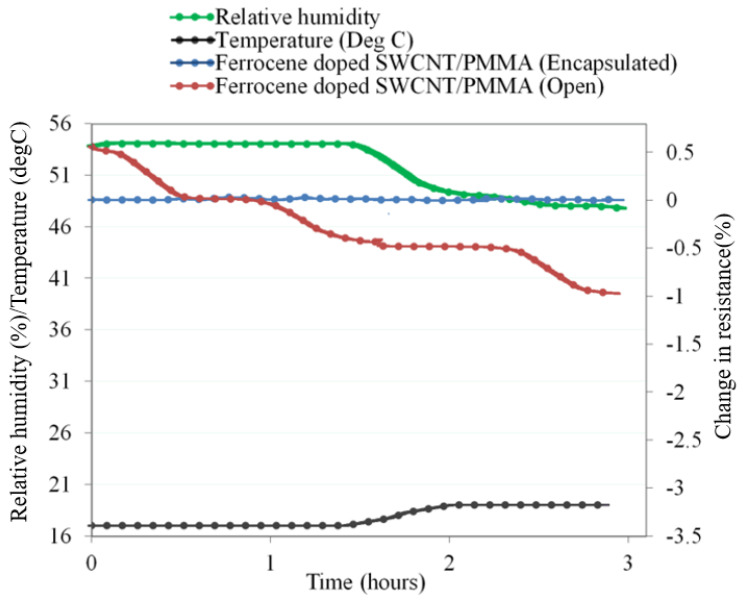
Ferrocene-doped SWCNT/PMMA thin-film device response to humidity in open and sealed devices. All data were recorded using an Arduino Nano microcontroller.

**Figure 10 nanomaterials-13-02653-f010:**
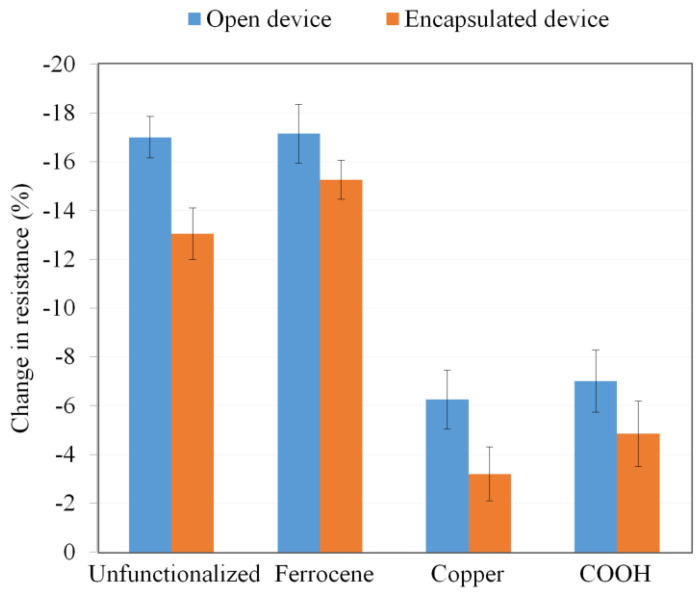
Unfunctionalized SWCNT/PMMA, ferrocene-doped SWCNT/PMMA, copper-coated SWCNT/PMMA, and carboxylic acid (COOH) functionalized SWCNT/PMMA thin-film device responses under open and encapsulated conditions to X-rays.

**Table 1 nanomaterials-13-02653-t001:** Initial resistance and change in resistance of device due to X-ray radiation.

Devices	Initial Resistance (kΩ)	Change in Resistance (%)
Ferrocene-SWCNT/PMMA	3.316 ± 1.69	(−)10 ± 3.32
Copper-coated-SWCNT/PMMA	1.636 ± 1.30	(−)4.7 ± 2.80
COOH-SWCNT/PMMA	2.576 ± 1.25	(−)4.5 ± 2.56

## Data Availability

The data presented in this study are available on request from the corresponding author.

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
