# Peer review of "Ionizing Radiation Sensing with Functionalized and Copper-Coated SWCNT/PMMA Thin Film Nanocomposites"

_nanomaterials, 2023, doi:10.3390/nano13192653_

Round 1
Reviewer 1 Report (Previous Reviewer 1)
The paper can be accepted in present form
Author Response
Reviewer 1 comments
Open Review
(x) I would not like to sign my review report
( ) I would like to sign my review report
Quality of English Language
( ) I am not qualified to assess the quality of English in this paper
( ) English very difficult to understand/incomprehensible
( ) Extensive editing of English language required
( ) Moderate editing of English language required
( ) Minor editing of English language required
(x) English language fine. No issues detected
|
Yes |
Can be improved |
Must be improved |
Not applicable |
|
|
Does the introduction provide sufficient background and include all relevant references? |
(x) |
( ) |
( ) |
( ) |
|
Are all the cited references relevant to the research? |
(x) |
( ) |
( ) |
( ) |
|
Is the research design appropriate? |
(x) |
( ) |
( ) |
( ) |
|
Are the methods adequately described? |
(x) |
( ) |
( ) |
( ) |
|
Are the results clearly presented? |
(x) |
( ) |
( ) |
( ) |
|
Are the conclusions supported by the results? |
(x) |
( ) |
( ) |
( ) |
Comments and Suggestions for Authors
The paper can be accepted in present form

Reviewer 2 Report (New Reviewer)
In this work, Guddi Suman et.al. prepared SWCNT/PMMA nanocomposites based thin films for ionizing radiation detection via reversible resistance variation. The results seem interesting. There are some issues remain and need to be well addressed before a reconsideration of this work. A major revision is suggested here.
1) The material characterizations were severely lacking and no information could be obtained about the homogeneity of the as-prepared films. As for the reproducibility of the experiment results, the authors should carefully treat this concern.
2) Provide more discussion about the merits of CNTs when detecting ion irradiation. Also, some relevant research about CNT-polymer based sensing is helpful to enrich the introduction part, see the reference Sensors and Actuators B 191 (2014) 24-30.
3) The reasons for the continuous decrease after radiation exposure are absent.
4) Add more discussion about the underlying detection mechanism by using the designed material system.
The quality of English language is fine.
Author Response
Reviewer 2 comments
Open Review
(x) I would not like to sign my review report
( ) I would like to sign my review report
Quality of English Language
( ) I am not qualified to assess the quality of English in this paper
( ) English very difficult to understand/incomprehensible
( ) Extensive editing of English language required
( ) Moderate editing of English language required
(x) Minor editing of English language required
( ) English language fine. No issues detected
|
Yes |
Can be improved |
Must be improved |
Not applicable |
|
|
Does the introduction provide sufficient background and include all relevant references? |
( ) |
(x) |
( ) |
( ) |
|
Are all the cited references relevant to the research? |
(x) |
( ) |
( ) |
( ) |
|
Is the research design appropriate? |
(x) |
( ) |
( ) |
( ) |
|
Are the methods adequately described? |
( ) |
(x) |
( ) |
( ) |
|
Are the results clearly presented? |
(x) |
( ) |
( ) |
( ) |
|
Are the conclusions supported by the results? |
( ) |
(x) |
( ) |
( ) |
Comments and Suggestions for Authors
In this work, Guddi Suman et.al. prepared SWCNT/PMMA nanocomposites based thin films for ionizing radiation detection via reversible resistance variation. The results seem interesting. There are some issues remain and need to be well addressed before a reconsideration of this work. A major revision is suggested here.
1) The material characterizations were severely lacking and no information could be obtained about the homogeneity of the as-prepared films. As for the reproducibility of the experiment results, the authors should carefully treat this concern.
Response: Added text line 196 paragraph 3, pg.6:
Homogeneity and adequate dispersion of the nanocomposites samples were confirmed by the SEM image of the ferrocene doped-SWCNT/PMMA, is shown in figure 1A, and the EDS of the Copper-coated nanotubes are shown in color mapping, which shows the distribution in supplemental S1 and of the ferrocene doped-SWCNT/PMMA in supplementary S2.
Following text is added in line 331, pg. 11.
These results show the reproducibility of the experimental results.
2) Provide more discussion about the merits of CNTs when detecting ion irradiation. Also, some relevant research about CNT-polymer based sensing is helpful to enrich the introduction part, see the reference Sensors and Actuators B 191 (2014) 24-30.
Response:
Following text is added to response this comment in line 42, pg. 2
Carbon nanotubes have also been found to be sensitive to low dose ion radiation resulting in vacancies [6].
The reference (Sensors and Actuators B 191 (2014) 24-30) is added in introduction section in pg. 2, ref [5].
3) The reasons for the continuous decrease after radiation exposure are absent.
Response: The following text is added in line 425, pg. 15, second paragraph.
]. It is noted in the data that there are instances where the resistance of the samples continues to decrease even after the X-rays were turned off (for example, the data in Figure 3 for each dose rate). This is likely due to relatively slow drift conduction to the CNT network of charge carriers generated in the polymer matrix (a poor conductor) during irradiation. Some charges likely become trapped and the thermal treatment discussed in Section 4.2 provides sufficient thermal energy to these trapped charges in the IDE oxide and polymer matrix so that the sample resistance relaxes to pre-irradiation condition in a relatively short time [28].
In addition, following sentence is added in second paragraph of line 279
Since the polymer is an insulator, the electrons take some time to drain from the polymer which is why the resistance continues to go down after the irradiation stops
4) Add more discussion about the underlying detection mechanism by using the designed material system.
Response:
The following Text in second paragraph of line 418, pg. 14 will answer Q.4
“The radiation induced charged carriers are transported to electrodes of the IDEs through the networking of nanotubes between the electrodes. Electrons transport contributes to conductivity, and holes are more likely to get trapped in the polymer matrix and the oxide layer of the IDE. This could explain the decrease in the resistance of nanocomposite thin-film device during X-ray exposure and the slow recovery of nanocomposite after X-rays are turned off [28, 30].”
The following Text in first paragraph of line 447, pg. 16 will answer Q.4
“The decrease in resistance of ferrocene-doped SWCNT was significantly higher than all other types of SWCNT studied at the lowest dose rate, as shown in Figure 6. The presence of ferrocene groups with SWCNT/PMMA increased the sensitivity to X-rays at lower dose rates, even in comparison with unfunctionalized, COOH functionalized and copper-coated SWCNT/PMMA devices. The bonds connecting ferrocene and CNT are broken during the X-ray exposure, which generated additional charge carriers for conduction.”

Reviewer 3 Report (New Reviewer)
Referee Report
On the paper “ Ionizing Radiation Sensing with Functionalized and Copper-Coated SWCNT/PMMA Thin Film Nanocomposites “ (nanomaterials-2623890) by the authors Guddi Suman, Merlyn Pulikkathara , Richard Wilkins, and LaRico Treadwell submitted to the Nanomaterials
This is interesting and useful paper. It reports ionizing radiation effects on functionalized single walled carbon nanotube (SWCNT)/ poly(methyl methacrylate) (PMMA) thin-film nanocomposites [SWNT/PMMA]. The nanocomposite was synthesized by the solution blending method and the resulting nanocomposite was spin-cast on interdigitated electrodes (IDEs). Carboxylic acid functionalized and copper-coated SWCNT/PMMA nanocomposite showed a reduced response to X-rays compared to unfunctionalized SWCNT/PMMA nanocomposite. Ferrocene-doped SWCNT showed a higher sensitivity to X-rays at lower dose rates. The result showed that a relatively economical, lightweight-designed prototype radiation sensor based on SWCNT/PMMA thin film devices could be produced by interfacing the devices with a modest microcontroller. The obtained experimental results are reliable without any doubts. However, I have some questions and additions. I would like to note a few points to improve the paper before it can be published:
1. The authors should in 1. Introduction and 2. Materials and Methods give examples of the formation of thin films:
(1). A.L. Kozlovskiy, M.V. Zdorovets, Synthesis, structural, strength and corrosion properties of thin films of the type CuX (X = Bi, Mg, Ni), J. Mater. Sci.: Mater. Electron. 30 (2019) 11819-11832. https://doi.org/10.1007/s10854-019-01556-x.
(2). A. Kotelnikova, T. Zubar, T. Vershinina, M. Panasiuk, O. Kanafyev, V. Fedkin, I. Kubasov, A. Turutin, S. Trukhanov, D. Tishkevich, V. Fedosyuk, A. Trukhanov, Saccharin adsorption influence on the NiFe alloy films growth mechanisms during electrodeposition, RSC Adv. 12 (2022) 35722–35729. https://doi.org/10.1039/D2RA07118E.
2. For metals and their alloys composite samples the stoichiometry is particularly important. The deviation from stoichiometry and appearance of the oxygen anions can lead to a change in the charge state of the cations, which in turn will greatly change the electronic parameters. That will seriously affect the practical application of the materials obtained. What is the oxygen stoichiometry of prepared samples? It is well known that the complex transition metal compounds easily allow the oxygen excess and/or deficit:
(3). S.V. Trukhanov, A.V. Trukhanov, A.N. Vasiliev, A.M. Balagurov, H. Szymczak, Magnetic state of the structural separated anion-deficient La0.70Sr0.30MnO2.85 manganite, J. Exp. Theor. Phys. 113 (2011) 819-825. https://doi.org/10.1134/S1063776111130127.
(4). A. Kozlovskiy, K. Egizbek, M.V. Zdorovets, M. Ibragimova, A. Shumskaya, A.A. Rogachev, Z.V. Ignatovich, K. Kadyrzhanov, Evaluation of the efficiency of detection and capture of manganese in aqueous solutions of FeCeOx nanocomposites doped with Nb2O5, Sensors 20 (2020) 4851. https://doi.org/10.3390/s20174851.
This should be discussed in 4. Discussions.
3. The authors should mention in 1. Introduction and 2. Materials and Methods such experimental methods of non-destructive testing and determination of microstresses in materials as X-ray or/and neutron diffraction:
(5). A.V. Trukhanov, L.V. Panina, S.V. Trukhanov, V.G. Kostishyn, V.A. Turchenko, D.A. Vinnik, T.I. Zubar, E.S. Yakovenko, L.Yu. Macuy, E.L. Trukhanova, Critical influence of different diamagnetic ions on electromagnetic properties of BaFe12O19, Ceram Int. 44 (2018) 13520-13529. https://doi.org/10.1016/j.ceramint.2018.04.183.
(6). D.I. Shlimas, A.L. Kozlovskiy, M.V. Zdorovets, Study of the formation effect of the cubic phase of LiTiO2 on the structural, optical, and mechanical properties of Li2±xTi1±xO3 ceramics with different contents of the X component, J. Mater. Sci.: Mater. Electron. 32 (2021) 7410-7422. https://doi.org/10.1007/s10854-021-05454-z.
4. The proposed 6 papers should be inserted in References.
The paper should be sent to me for the second analysis after the major revisions.

Minor editing of English language required
Author Response
Reviewer 3 comments
Referee Report
On the paper “ Ionizing Radiation Sensing with Functionalized and Copper-Coated SWCNT/PMMA Thin Film Nanocomposites “ (nanomaterials-2623890) by the authors Guddi Suman, Merlyn Pulikkathara , Richard Wilkins, and LaRico Treadwell submitted to the Nanomaterials
This is interesting and useful paper. It reports ionizing radiation effects on functionalized single walled carbon nanotube (SWCNT)/ poly(methyl methacrylate) (PMMA) thin-film nanocomposites [SWNT/PMMA]. The nanocomposite was synthesized by the solution blending method and the resulting nanocomposite was spin-cast on interdigitated electrodes (IDEs). Carboxylic acid functionalized and copper-coated SWCNT/PMMA nanocomposite showed a reduced response to X-rays compared to unfunctionalized SWCNT/PMMA nanocomposite. Ferrocene-doped SWCNT showed a higher sensitivity to X-rays at lower dose rates. The result showed that a relatively economical, lightweight-designed prototype radiation sensor based on SWCNT/PMMA thin film devices could be produced by interfacing the devices with a modest microcontroller. The obtained experimental results are reliable without any doubts. However, I have some questions and additions. I would like to note a few points to improve the paper before it can be published:
- The authors should in 1. Introduction and 2. Materials and Methods give examples of the formation of thin films:
(1). A.L. Kozlovskiy, M.V. Zdorovets, Synthesis, structural, strength and corrosion properties of thin films of the type CuX (X = Bi, Mg, Ni), J. Mater. Sci.: Mater. Electron. 30 (2019) 11819-11832. https://doi.org/10.1007/s10854-019-01556-x.
(2). A. Kotelnikova, T. Zubar, T. Vershinina, M. Panasiuk, O. Kanafyev, V. Fedkin, I. Kubasov, A. Turutin, S. Trukhanov, D. Tishkevich, V. Fedosyuk, A. Trukhanov, Saccharin adsorption influence on the NiFe alloy films growth mechanisms during electrodeposition, RSC Adv. 12 (2022) 35722–35729. https://doi.org/10.1039/D2RA07118E.
Response:
References [1] and [2] have been included in [20] and [21] and following text have been in line 52;
Thin films has been a way of creating sensors and have been studied in literature for example; the thin films of the type CuX (X = Bi, Mg, Ni) studied the impact of the concentration of copper and phases containing copper on degradation and oxidation of thin films [18]. Another example of synthesis of NiFe films using direct, pulse, and pulse-reverse electrodeposition modes studied the deposit composition, crystal structure, and surface microstructure [19]. There has been mechanical modeling of thin films [20, 21]
- For metals and their alloys composite samples the stoichiometry is particularly important. The deviation from stoichiometry and appearance of the oxygen anions can lead to a change in the charge state of the cations, which in turn will greatly change the electronic parameters. That will seriously affect the practical application of the materials obtained. What is the oxygen stoichiometry of prepared samples? It is well known that the complex transition metal compounds easily allow the oxygen excess and/or deficit:
(3). S.V. Trukhanov, A.V. Trukhanov, A.N. Vasiliev, A.M. Balagurov, H. Szymczak, Magnetic state of the structural separated anion-deficient La0.70Sr0.30MnO2.85 manganite, J. Exp. Theor. Phys. 113 (2011) 819-825. https://doi.org/10.1134/S1063776111130127.
(4). A. Kozlovskiy, K. Egizbek, M.V. Zdorovets, M. Ibragimova, A. Shumskaya, A.A. Rogachev, Z.V. Ignatovich, K. Kadyrzhanov, Evaluation of the efficiency of detection and capture of manganese in aqueous solutions of FeCeOx nanocomposites doped with Nb2O5, Sensors 20 (2020) 4851. https://doi.org/10.3390/s20174851.
This should be discussed in 4. Discussions. 2
Response:
Following sentence is added in response in line 444 and given references have been included in manuscript as [46] and [47]
The complex transition metal compounds easily causes the oxygen excess and/or deficit [44, 45]. In addition, PMMA matrix itself has a stoichiometric formula (C5O2H8)n, and it breaks down to monomers at 1650C causing oxygen diffusion [46, 47].
- The authors should mention in 1. Introduction and 2. Materials and Methods such experimental methods of non-destructive testing and determination of microstresses in materials as X-ray or/and neutron diffraction:
(5). A.V. Trukhanov, L.V. Panina, S.V. Trukhanov, V.G. Kostishyn, V.A. Turchenko, D.A. Vinnik, T.I. Zubar, E.S. Yakovenko, L.Yu. Macuy, E.L. Trukhanova, Critical influence of different diamagnetic ions on electromagnetic properties of BaFe12O19, Ceram Int. 44 (2018) 13520-13529. https://doi.org/10.1016/j.ceramint.2018.04.183.
(6). D.I. Shlimas, A.L. Kozlovskiy, M.V. Zdorovets, Study of the formation effect of the cubic phase of LiTiO2 on the structural, optical, and mechanical properties of Li2±xTi1±xO3 ceramics with different contents of the X component, J. Mater. Sci.: Mater. Electron. 32 (2021) 7410-7422. https://doi.org/10.1007/s10854-021-05454-z.
Response:
Above references have been added references as [20] and [24] and following text have been added in lines 57
The experimental thermomechanical property was studied on materials such as ceramics [24],
following text have been added in lines 57
There has been mechanical modeling of thin films [20].
- The proposed 6 papers should be inserted in References.
The paper should be sent to me for the second analysis after the major revisions.
Response:
The proposed 6 papers have been inserted in References.

Round 2
Reviewer 2 Report (New Reviewer)
The authors have carefully responded to all my concerns. Thus, the current manuscript deserves a publication approval in this journal.
Reviewer 3 Report (New Reviewer)
Referee Report
On the paper “ Ionizing Radiation Sensing with Functionalized and Copper-Coated SWCNT/PMMA Thin Film Nanocomposites “ (nanomaterials-2623890-v2) by the authors Guddi Suman, Merlyn Pulikkathara , Richard Wilkins, and LaRico Treadwell submitted to the Nanomaterials
This paper has been well corrected and it can be recommended.

Minor editing of English language required
This manuscript is a resubmission of an earlier submission. The following is a list of the peer review reports and author responses from that submission.
Round 1
Reviewer 1 Report
The manuscrit entitled: "Ionizing Radiation Sensing with Functionalized and Copper-Coated SWCNT/PMMA Thin Film Nanocomposites" is a work dealing with the reaction after specific Xray radiation exposure of MWCNTs and SWCNT PMMA thin film nanocomposites. From a first approach, this is a rather rational work given the fact the PMMA is a polymer that is quite vulnerable to X-rays and as an effect the ageing of PMMA is high. Having that said the authors have to explain
- why did they choose PMMA over other polymers that are stable in Xray beam?
- what is the life expected of the device - a sensor that is created?
- are there possible to have an application of the composite as a sensor?
- is there a maximum loading of CNTs identified that is affecting the stability of the PMMA?
- what can the authors comment on the long term stability of such composites based on CNTs/PMMA?
Reviewer 2 Report
In this manuscript, the authors studied the ionizing radiation effects of single walled carbon nanotube (SWCNT)-based nanocomposite films [Ferrocene-doped SWCNT/PMMA, Copper-coated SWCNT/PMMA, and Carboxylic acid (COOH) functionalized SWCNT/PMMA)]. They observed the resistance changes of those composite films with X-ray irradiation and suggested the application as a radiation sensor in the future.
In this study, the authors only focused on the electrical measurement of the fabricated devices without material characterization. Because the authors used surface functionalized or metal coated nanomaterials, material characterization should be required to confirm the surface condition of nanomaterials such as XPS and FT-IR etc. In addition, the authors described their results in the Discussion part with several references, but the evidence seems to be insufficient to support their assumption. The additional schematic diagram and analyses should be required. After major revision, the manuscript could be published in “Nanomaterials”.
1. The authors used four different types of materials [unfunctionalized SWCNT/PMMA, Ferrocene-doped SWCNT/PMMA, Copper-coated SWCNT/PMMA, and Carboxylic acid (COOH) functionalized SWCNT/PMMA)] in this manuscript. The authors should provide basic surface information for those materials through surface analyses such as XPS and FT-IR.
2. Regarding the measured devices, the authors used interdigitated electrodes and network-type SWCNT composite, and it can induce inhomogeneous electrical properties of the sensor devices. Therefore, the statistical data of the fabricated devices should be added in the manuscript such as average values of initial resistance or resistance change for the composite thin films.
3. In 1st and 2nd paragraphs of Discussion section, the authors described the resistance changes in different types of composite films depending on the applied X-ray dose rates. If the authors add schematic diagram, it would be very helpful for the readers.
4. In 2nd paragraph of Discussion section, the authors said that SWCNTs could induce the reduction of oxidation of the copper. To prove this, the additional analyses such as XPS should be required.
5. Regarding the relative humidity test in encapsulated condition, even though this study is about the functionalized and copper-coated SWCNT/PMMA thin film nanocomposites, the authors only provided the experimental data for unfunctionalized SWCNT/PMMA nanocomposite as shown in Figure 9 and 10. The additional measurement using other types of nanocomposites [Ferrocene-doped SWCNT/PMMA, Copper-coated SWCNT/PMMA, and Carboxylic acid (COOH) functionalized SWCNT/PMMA)] should be required.